

# Bilateral asymmetry of skin temperature is not related to bilateral asymmetry of crank torque during an incremental cycling exercise to exhaustion

Athos Trecroci[1], Damiano Formenti[1], Nicola Ludwig[2], Marco Gargano[2], Andrea Bosio[3], Ermanno Rampinini[3] and Giampietro Alberti[1]

[1] Department of Biomedical Sciences for Health, Università degli Studi di Milano, Milan, Italy
[2] Department of Physics, Università degli Studi di Milano, Milan, Italy
[3] Human Performance Laboratory, MAPEI Sport Research Centre, Varese, Italy

## ABSTRACT

Although moderate relationships ($|r| \sim 0.5$) were reported between skin temperature and performance-related variables (e.g., kinetic), it remains unclear whether skin temperature asymmetry reflects muscle force imbalance in cycling. Therefore, the aim of this study was to assess whether a relationship exists between kinetic and thermal asymmetry during a fatiguing exercise. Ten elite cyclists were enrolled and tested on a maximal incremental cycling test. Peak crank torques of both legs were obtained at the initial and final workload. Likewise, bilateral skin temperatures were recorded before and after exercise. Asymmetric indexes were also calculated for kinetic ($AI_K$) and skin temperature ($AI_T$) outcomes. The bilateral peak crank torques showed a larger difference at the final compared to the initial workload ($p < 0.05$) of the incremental exercise. Conversely, the bilateral skin temperature did not show any differences at both initial and final workload ($p > 0.05$). Additionally, trivial relationships were reported between $AI_K$ and $AI_T$ ($-0.3 < r < 0.2$) at the initial and final workload. The obtained results showed that changes in bilateral kinetic values did not reflect concurrent changes in bilateral skin temperatures. This finding emphasizes the difficulty of associating the asymmetry of skin temperature with those of muscle effort in elite cyclists. Lastly, our study also provided further insights on thermal skin responses during exhaustive cycling exercise in very highly-trained athletes.

Corresponding author
Athos Trecroci,
athos.trecroci@unimi.it

## INTRODUCTION

The assessment of bilateral asymmetry assumes a relevance to determine potential risks of injury and functional deficits of unhealthy athletes and non-athletes (*Rodrìguez-Sanz et al., 2017*; *Schmitt, Paterno & Hewett, 2012*; *Zwolski et al., 2015*) as well as to quantify motor performance of healthy and athlete peers (*Carabello et al., 2010*; *Liu & Jensen, 2012*; *Vardasca et al., 2012*; *Zaproudina et al., 2008*; *Bishop, Turner & Read, 2017*).

In cycling, lower-limb symmetry refers to a balance in the force production of a leg compared to the contralateral within repeated pedal cycles (*Sanderson, 1990*). Although

pedaling represents a symmetrical motor task, it is common for cyclists to experience imbalances in the force applied to the crank between right and left limbs (*Sanderson, 1990*). It has been claimed that an existing imbalance between the pedal forces would limit cycling performance due to an early onset of fatigue (*Carpes et al., 2011*; *Bini et al., 2007*; *Bini et al., 2016*; *Bertucci, Arfaoui & Polidori, 2012*). Unfortunately, only a few studies have addressed bilateral asymmetry in cycling reporting differences in the kinetic variables (e.g., crank torques and pedal-to-crank angles) (*Bini & Hume, 2015*; *Carpes et al., 2007*).

Recently, a moderate relationship between muscle activation (using surface electromyography) and skin temperature (using infrared thermography, IRT) (e.g., $r < -0.5$) has been found in active subjects during an incremental cycling test. Temperature outcomes were obtained before, immediately after and 10 min after the test from four specific body regions (i.e., rectus femoris, vastus lateralis, biceps femoris, and gastrocnemius medialis) (*Priego Quesada et al., 2015a*). The authors reported a significant moderate inverse relationship (i.e., $r = -0.58$) between skin temperature changes and neuromuscular activity in the vastus lateralis, which markedly contributes to force production (*Priego Quesada et al., 2015a*). Hence, participants with greater muscle activation exhibited a lower increase in skin temperature at the end of the exercise. Additionally, another recent study reported significant correlation between skin temperature changes and muscle power output during an isokinetic leg extension activity on quadriceps muscle (*Hadžić et al., 2015*). Likewise, *Priego Quesada et al. (2017)* found a moderate (i.e., $r = 0.5$) correlation between skin temperature and peak power output during an incremental cycling test to exhaustion on lower limb muscles. However, correlational analysis should be interpreted with caution because it does not imply a cause–effect relationship.

The use of IRT has also been proposed to indirectly assess muscle damage and inflammation following exertion via skin temperature assessment (*Hildebrandt, Raschner & Ammer, 2010*). Accordingly, a recent study found significant correlations of skin temperature of right and left lower limbs with creatine kinase activity after two consecutive matches in soccer players (*De Andrade Fernandes et al., 2017*). The rationale was that micro-injuries caused by bilateral muscle strain in response to fatiguing exercise would have reflected changes in skin temperature asymmetry (*Ring & Ammer, 2012*). Hence, it would be reasonable to consider potential changes in skin temperature as a result of such muscle strain (e.g., due to a maximal incremental workload) in cycling. Therefore, determining factors that influence the thermal asymmetry during an incremental cycling may be valuable for monitoring state of fatigue and injury risk, and bilateral peak crank torque could be one possible candidate.

To the best of our knowledge, no studies have addressed the relationship between bilateral asymmetry of crank torque and skin temperature in cyclists. It has been recently found that skin temperature was primarily affected by subcutaneous tissues and physical fitness level (*Priego Quesada et al., 2017*). Accordingly, *Priego Quesada et al. (2017)* found a negative correlation of skin temperature with body fat (i.e., $r = -0.5$) and a moderate association of skin temperature with peak power (i.e., $r = 0.5$) in response to an incremental cycling test to exhaustion. Hence, the use of skin temperature as an index of muscle force imbalance remains uncertain. The main issue refers to a large number of environmental

(e.g., humidity), individual (e.g., subcutaneous tissue) and technical (e.g., protocol) factors affecting skin temperature variation/assessment (*Fernàndez-Cuevas et al., 2015*) in response to exercise. All together, these findings seem not sufficient to expect that asymmetries in skin temperature are related to asymmetries in crank torque. However, new perspectives into the assessment of asymmetry are required to add physiological information to muscle force imbalance, especially concerning highly-trained athletes. For instance, elite cyclists exhibit lower body fat levels compared to less trained peers (i.e., club cyclists, recreational and non-cyclists). Since body fat is one of the variables affecting skin temperature, studying highly-trained cyclists (with lower body fat percentage) may provide a clearer picture of the skin temperature behavior, in terms of bilateral asymmetry throughout exercise.

Therefore, the aims of the present study were twofold: (a) to investigate the presence of skin temperature and kinetic asymmetry (i.e., peak crank torque) in response to maximal incremental cycling exercise in elite cyclists; (b) to assess whether a relationship exists between both kinetic and thermal asymmetric indexes at the initial and final workload of the maximal incremental cycling exercise. Given the magnitude of the relationships (moderate) previously reported between performance-related variables and skin temperature (combined with the main role of body fat), we hypothesized that the bilateral differences in the peak crank torque may not be associated with the bilateral differences in skin temperature, thus adding extended knowledge on determinants of the thermal imbalance in highly-trained cyclists.

## MATERIAL AND METHODS

### Participants

Ten male elite cyclists participated voluntarily in the study. Their mean and standard deviation values for age, body mass, height, body fat percentage, maximal oxygen uptake, and power output were: $21.4 \pm 2.6$ years, $68.9 \pm 6.8$ kg, $1.76 \pm 0.39$ m, $8.9 \pm 2.4\%$, $67.6 \pm 5.4$ ml/kg/min, $436 \pm 43$ W. According to the declaration of Helsinki, the study was approved by the Ethical Committee of the Università degli Studi di Milano (Approval number: 2/12). A written informed consent including a complete description of procedures and potential risks was provided, and signed by all the participants. They were also instructed to avoid high-intensity or strenuous physical activity 24-h prior to testing.

### Procedures

All participants carried out a single experimental session at the same time of day (9.00 a.m.–12.00 a.m.) (*Marins et al., 2015*). They were asked to refrain from strenuous exercise as well as consuming caffeine, drugs or other medications with potential effect on cardiovascular and thermoregulatory functions at least 48 h prior to testing. Additionally, they followed a standardized light meal 2–3 h prior to testing. In each session, prior to testing, stature and body weight were recorded by a stadiometer and weighing scale to both the nearest 0.1 cm and 0.1 kg, respectively. Moreover, it was ensured that the thighs skin of each participant were hairless and devoid of cosmetics products following the reference standard protocols (*Fernàndez-Cuevas et al., 2015*). Body fat percentage was estimated using the equation proposed by Jackson and Pollock (*Jackson & Pollock, 1978*). Afterwards, participants rested

on an isolating mat positioned on the floor, without touching their legs, for a period of 10 min. This procedure was employed to acclimatise the body skin of each cyclist to the environmental condition of laboratory (temperature 22–23 °C; relative humidity 50 ± 5%; constant natural and fluorescent lighting and no direct ventilation) (*Marins et al., 2014*). After the acclimatization period, they were instructed to mount on the cycle ergometer. A customised setting was used for each participant to replicate the position assumed on his own bike. Then, all thermal acquisitions were performed asking the cyclists not to pedal and staying upright with the leg extended toward the floor.

The asymmetric index was considered to quantify the between-limb difference at initial and final workload respectively for both kinetic and skin temperature values. To calculate the asymmetric index related to the kinetic values ($AI_K$), we used the method proposed by *Impellizzeri et al. (2007)* based on the following equation: (stronger–weaker)/stronger × 100. The stronger leg was defined as the lower limb eliciting the highest value of torque during the propulsion phase. Vice versa, the weaker leg was defined as the lower limb with the lowest value of torque during the propulsion phase. A negative sign (−) was arbitrary attributed when the stronger leg was the left one, while a positive sign (+) was assigned when the stronger leg was the right one. The same procedures were implemented to obtain the asymmetric index related to the bilateral skin temperatures ($AI_T$). To calculate the $AI_T$, the adjectives "stronger" and "weaker" were replaced as "warmer" and "colder", respectively, and their data referred exclusively to the starting and exhaustion time points. Thus, a negative sign (−) was arbitrary attributed when the warmer leg was the left one, while a positive sign (+) was assigned when the warmer leg was the right one.

## Incremental cycling test

After 10 min of warm up performed with a constant load of 100 W, participants completed an incremental maximal cycling test to determine maximal oxygen uptake (breath-by-breath VMX 29 system; Sensormedics, Yorba Linda, CA, USA) and maximal power output. Each participant started at a workload of 100 W with an increase of 25 W every minute until exhaustion. They remained seated throughout the whole cycling test. Pedaling cadence was kept constant throughout the test in a range of 90 ± 3 rpm. Time to exhaustion point corresponded with the cyclist's incapacity to maintain a cadence above 87 rpm. Maximal power output was considered as the workload value of the last stage fully concluded.

## Kinetic variables

Independent pedal force outcomes for both right and left cranks were constantly measured by a built-in modified strain gauge technology mounted on the cycle ergometer (Lode Excalibur Sport; Lode B.V., Groningen, Netherlands). The force measurements occurred with a rotational resolution of 2° and an accuracy of 0.5 N. Peak crank torque was recorded during the propulsion phase, which was defined as the set of crank torques applied from 0° to 180° of a pedal cycle in clockwise direction. The overall torque values were collected during the last thirty pedal cycles of the first workload (i.e., 100 W) and the last completed workload (i.e., exhaustion) of the maximal incremental test.

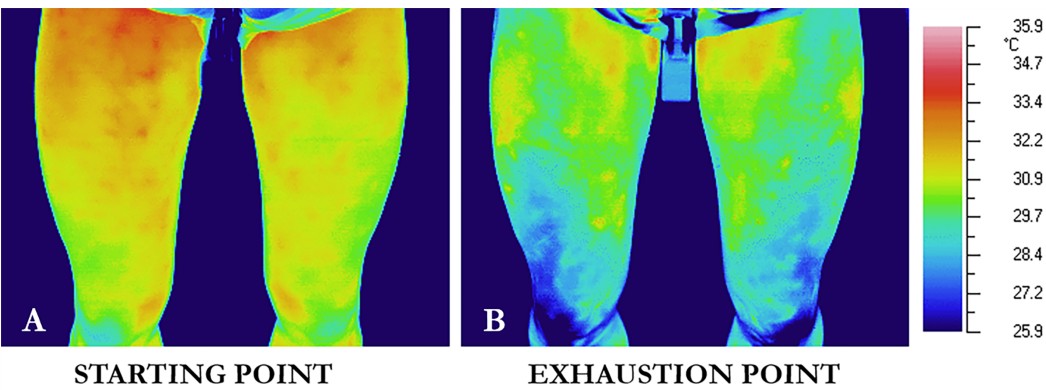

STARTING POINT        EXHAUSTION POINT

**Figure 1** **Thermal images of a representative participant at the initiation of the exercise (starting point, A), and immediately after the cessation of the exercise (exhaustion point, B).**

### Skin temperature

Skin temperature of muscle quadriceps was measured contactless using infrared thermal imaging. Thermal images arising from the anterior surface of the participants' thighs were recorded by a 14-bit digital infrared thermal camera (AVIO, TVS-700, $320 \times 240$ Microbolometric Array, 8–14 μm spectral range, NETD 0.07 K, 35 mm lens; Nippon Avionics Co, Ltd, Tokyo, Japan). The thermal camera was placed directly in front of the cycle ergometer at the fixed height of 118 cm with a background at constant temperature. Recordings were implemented using a digital frame grabber setting to capture one image per 10 s. Two thermal images were captured at each specific time point: basal (before the warm up), starting point (after the warm up), exhaustion point, and post-exercise period (3 min and 6 min after the exhaustion). Emissivity value was set to 0.98.

The thermal images were analyzed with a dedicated software for thermal image processing (GRAYESS® IRT Analyzer, Version 4.8; GRAYESS, Bradenton, FL, USA) by the Tmax method, which was proposed by *Ludwig et al. (2014)* and utilised in a recent study to evaluate skin temperature dynamic of thighs surfaces in highly-trained cyclists during exercise (*Ludwig et al., 2016*). A region of interest (ROI) per each thigh was selected by an operator following physical location based on classical definition of body regions (Fig. 1) (*Formenti et al., 2017*). No landmarks were used to select the interested region. Within such region, the software automatically selected the five hottest pixels having a minimum distance of five pixels from each other. Tmax was then obtained by averaging the area of $5 \times 5$ pixels around the hottest pixels. This value was therefore representative of an amount of 125 pixels. Finally, the skin temperature values of the left and right region of interest were averaged, yielding one representative temperature value for each time point (*Formenti et al., 2016*).

### Statistical analyses

The assumption of normality was verified by the Shapiro Wilk's test for each variable. A two-way repeated measures analysis of variance (ANOVA RM) was used to compare

**Table 1** $AI_K$ and $AI_T$ values at initial and final workload together with Pearson's product-moment correlations of cyclists ($N = 10$).

| Workload | $AI_T$ (%) Mean $\pm$ SD | $AI_K$ (%) Mean $\pm$ SD | $r$ [95% CI] | Effect size |
|---|---|---|---|---|
| Initial (100 W) | $-0.32 \pm 1.38$ | $-2.12 \pm 9.05$ | $-0.277$ [$-0.77$, 0.43] | Trivial |
| Final (exhaustion) | $-0.47 \pm 1.94$ | $5.61 \pm 6.62$ | 0.162 [$-0.52$, 0.72] | Trivial |

Notes.

$AI_T$, bilateral asymmetric index of skin temperature; $AI_K$, bilateral asymmetric index of peak crank torque; $r$, Pearson correlation coefficient.

skin temperature and kinetic values separately for the propulsion phase of right and left lower limbs across the time points (within-subjects factor). Least Significant Difference (LSD) test was used for pairwise multiple comparisons among the time points (e.g., basal, starting point, exhaustion, after 3 min and 6 min). As a measure of effect size for ANOVA partial eta squared ($p\eta^2$) was reported (*Richardson, 2011*). The thresholds for a small, moderate and a large effect were defined as 0.01, 0.06, and 0.14, respectively (*Cohen, 1988*). An alpha threshold of $p < 0.05$ was set to identify statistical significance with a desire power of 80% ($\beta$ error of 0.2). The relationship between the two asymmetric indexes ($AI_K$ and $AI_T$) was assessed by Pearson's product-moment correlation at initial and final workload, respectively. The correlation coefficients were interpreted using the magnitude scale proposed by *Hopkins et al. (2009)* as follows: <0.1, trivial; 0.1–0.3, small; 0.3–0.5, moderate; 0.5–0.7, large; 0.7–0.9, very large; >0.9, nearly perfect. Statistical analysis was performed using the IBM SPSS Statistics software (v. 21; New York, NY, USA). Data are reported as mean $\pm$ SD.

## RESULTS

The two-way ANOVA RM revealed a statistically significant interaction between limbs in the peak crank torques, ($F_{1,9} = 14.813$, $p = 0.004$, $p\eta^2 = 0.622$), indicating that the bilateral force applied to the crank was not symmetrical at the final workload. The overall kinetic data are shown in Fig. 2. Figure 3 shows the skin temperature responses of right and left thighs in correspondence to the selected time points before and after the cycling test. As regards skin temperature, the two-way ANOVA RM showed non-significant interaction between thighs ($F_{4,36} = 0.427$, $p = 0.643$, $p\eta^2 = 0.045$) across the selected time points, suggesting that right-to-left temperature values did not significantly differ to each other throughout the exercise (basal, starting point, exhaustion, after 3 min and after 6 min). The main effect of limbs was not significant ($F_{1,9} = 1.717$, $p = 0.223$, $p\eta^2 = 0.160$), whereas the main effect of time was significant ($F_{4,36} = 10.504$, $p < 0.0001$, $p\eta^2 = 0.539$). In accordance, LSD pairwise multiple comparisons revealed a significant decrease in skin temperature of both thighs at the exhaustion point compared to those recorded before the exercise (basal, $p = 0.008$; starting point, $p = 0.001$) and within the recovery period (after 3 min, $p < 0.0001$; after 6 min, $p = 0.001$).

$AI_K$ and $AI_T$ values at initial and final workload together with Pearson's product-moment correlations are shown in Table 1. Trivial and not significant correlations (Table 1) were found between $AI_K$ and $AI_T$ at both initial ($p = 0.438$) and final workload ($p = 0.654$).

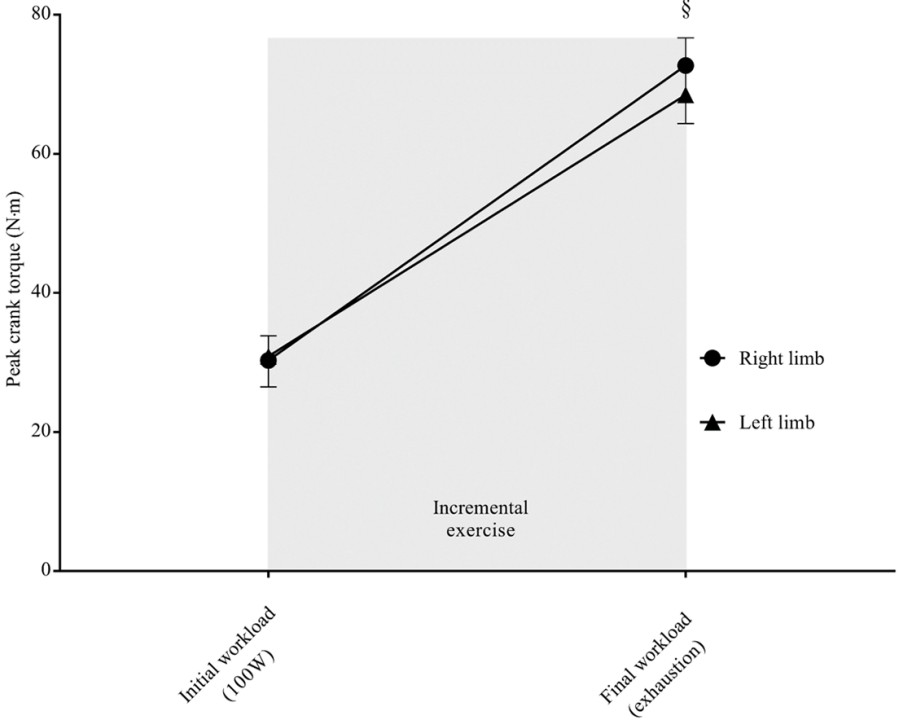

**Figure 2  Peak crank torque values at the initial and final workload.** §Significant difference between right and left limb ($p < 0.05$).

## DISCUSSION

The main findings of the present study were twofold: (a) the two-way ANOVA RM revealed a statistically significant interaction for peak crank torques, but not for skin temperatures between right and left limbs; (b) although positive $AI_K$ and negative $AI_T$ indicated that the right limb was the stronger as well as the colder, non significant correlations were found between the two asymmetric indexes at final workload. All together, these findings contribute to reinforce our hypothesis based on inconsistence relationship between bilateral differences in peak crank torque and skin temperature.

To the extent of our knowledge, this is the first study that sought to compare kinetic variables and skin temperature of right and left limbs in response to an incremental maximal exercise in highly-trained cyclists. Two previous studies have investigated skin temperature asymmetry of calves' regions during a submaximal (i.e., 100–250 W) cycling performance in master athletes (*Arfaoui et al., 2014*; *Bertucci, Arfaoui & Polidori, 2012*). In the study of *Bertucci, Arfaoui & Polidori (2012)*, eleven trained master cyclists cycled at 100 W for 10 min, and then the intensity was increased by 50 W every 3 min up to 200 W, and after that, there was set a last step at 250 W for further 2 min. The results showed significant bilateral differences in the crank torque values at 150, 200, and 250 W in favor of the right limb. Afterwards, *Arfaoui et al. (2014)* replicated the study proposed by *Bertucci, Arfaoui & Polidori (2012)* to assess such bilateral differences in the force applied to the crank using

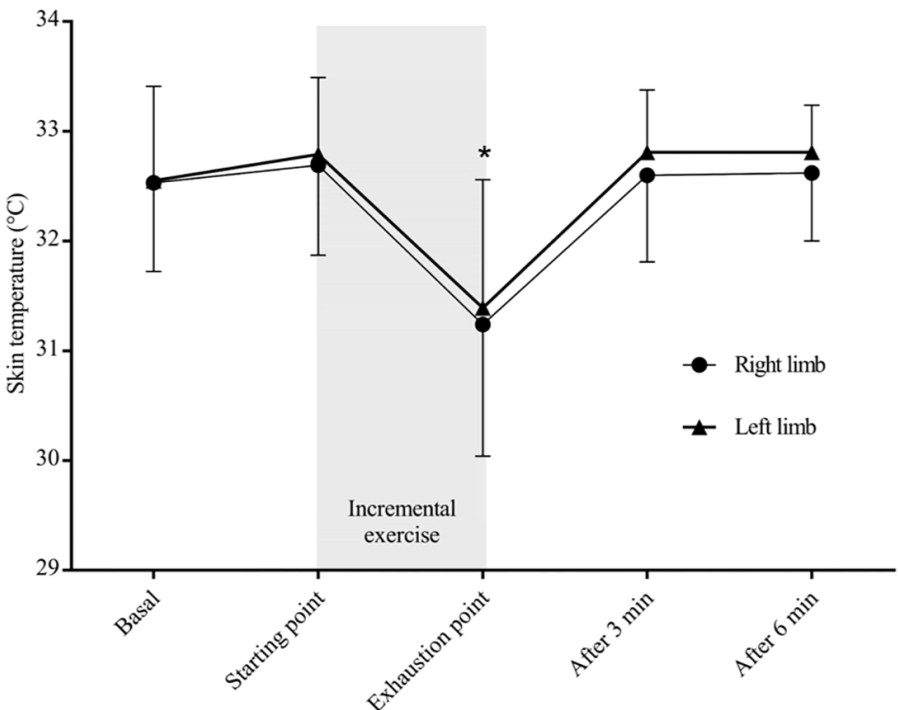

**Figure 3** **Skin temperature response of right and left limbs across time points.** *Significant difference compared to all of the time points ($p < 0.05$).

IRT analysis (i.e., monitoring skin temperature). Hence, *Arfaoui et al. (2014)* tested the same eleven master cyclists recruited by *Bertucci, Arfaoui & Polidori (2012)* on the identical protocol. Thermal images were recorded before, during, and at the end of the exercise on both calves. As a result, with respect to the findings (i.e., presence of asymmetry in the kinetic variable) showed by *Bertucci, Arfaoui & Polidori (2012)*, no significant differences were found between bilateral skin temperatures, showing a parallel decrement in the skin temperatures of both calves throughout the exercise.

The present findings seem to be in line with those observed by *Bertucci, Arfaoui & Polidori (2012)* and *Arfaoui et al. (2014)*. Indeed, although they involved an incremental submaximal rather than maximal workloads, bilateral skin temperature did not reflect the changes occurred in the bilateral peak crank torques in well-trained cyclists (i.e., masters), as also reported for the present highly-trained cyclists (i.e., elite). This suggests that the use of IRT would add physiological information (e.g., skin temperature profile during an incremental workload in highly-trained counterparts) that are not associated to force asymmetry during a maximal incremental exercise in elite cyclists.

Therefore, skin temperature asymmetry as index of muscle force imbalance seems to be inappropriate. Interestingly, *Chudecka et al. (2015)* sought to evaluate bilateral changes in skin temperature, by IRT, after symmetrical and asymmetrical sport activities in professional rowers and handball players. Bilateral skin temperatures of symmetrical front and back sides of arm, forearm, thigh, and trunk were recorded using an infrared thermal

camera before and after the exercise. A significant decrease in mean skin temperature immediately after the exercise was observed in both groups of athletes. Significant bilateral differences in skin temperatures of the body regions were observed only in handball players. Conversely, rowers did not show differences between right and left sides of each body region. Although *Chudecka et al. (2015)* involved different exercise tasks rather than cycling, their results appeared to support the use of IRT for assessing bilateral skin temperature changes through asymmetrical (i.e., handball) rather than symmetrical exercise (e.g., rowing). Compared to symmetrical activities, those asymmetrical may determine a predominant involvement of specific body regions (dominant body-side), which in turn elicits larger increases of heat (to dissipate) against less involved body regions, thus affecting differently skin temperature.

Accordingly, in the current study, along with that of *Arfaoui et al. (2014)*, no bilateral skin temperature differences were observed on cycling performance (i.e., symmetrical task), regardless of exercise intensity. Reasonably, the nature of the task falls among the factors that contribute to influence skin temperature (e.g., body fat, humidity, skin wetness, etc...), which explain the trivial relationship ($-0.3 < r < 0.2$) between the two asymmetric indexes reported in the present results.

A further aspect that should be considered refers to the high-training level of the present cyclists that might have allowed them to exhibit lower mean values of $AI_K$ (Table 1) compared to the normative threshold ($AI = 10\%$) beyond that asymmetry is considered abnormal (*Carpes, Mota & Faria, 2010*). Indeed, being related to performance changes (*Bini & Hume, 2015*), bilateral asymmetry might be more pronounced in low-trained athletes (i.e., sub-elite or recreational cyclists) with a substantial implication on bilateral skin temperature difference. Nevertheless, this remains speculative. Indeed, due to paucity of research studies assessing muscle activation, caution should be used when drawing conclusions on a possible relation between bilateral changes in skin temperature and muscle effort.

The clear fall in skin temperature observed throughout the current incremental exercise was in accordance with the findings of *Merla et al. (2010)*. In that study, the authors investigated changes in the skin temperature of anterior thighs of well-trained participants during an incremental running test. Skin temperature (recorded by IRT) decreased as the participants began the exercise, whereas a further skin temperature decrement occurred throughout the exercise, until the exhaustion.

As confirmed by *Merla et al. (2010)*, the skin temperature decrement observed in our study may be due to the continuous vasoconstrictor response, attributable to an increase in catecholamine and other vasoconstrictor hormones released with the increasing exercise intensity (*Vainer, 2005*). It can be assumed that a continuous request of blood flow to the active muscles by incremental workloads would contribute to a continuous skin temperature reduction. However, this should be attributed not only to a decrease in skin blood flow, but may be also related to a small amount of sweat production, even it seems to have a secondary role in decreasing skin temperature (*Priego Quesada et al., 2015a*; *Priego Quesada et al., 2015b*).

### Limitations

The current study presents some weaknesses that should be acknowledged. First, the adopted experimental setting is far from regular training or competition environments, thus related changes in the exercise intensity, position (i.e., sitting and standing postures) (*Chen et al., 2016*) as well as in the climate conditions may also affect both kinetic and skin temperature values, and their relationship in a different manner. Second, this study did not consider pedal power measurement, which would have provided additional and extended information on muscle effort asymmetry. Along with this, musculoskeletal modeling would also have been a further option for assessing muscle asymmetries in term of biomechanical simulation. Third, since numerous lower limb muscles contribute to force production in cycling, the thermal investigation of a single muscle region (i.e., quadriceps) may represent a limit. Further studies should design experimental settings with an appropriate number of infrared thermal cameras to record the skin temperature changes of all muscles involved in cycling.

## CONCLUSIONS

In conclusion, the present study showed that bilateral differences in the kinetic variables did not reflect concurrent bilateral differences in the skin temperature during a maximal incremental cycling test. This finding emphasizes the difficulty of associating skin temperature with muscle cycling effort, which is also reinforced by the trivial relationship between kinetic and skin temperature asymmetric indexes.

## ACKNOWLEDGEMENTS

Thanks to Alessio Rossi PhD for his technical support.

### Funding

The authors received no funding for this work.

### Competing Interests

The authors declare there are no competing interests.

### Author Contributions

- Athos Trecroci conceived and designed the experiments, performed the experiments, analyzed the data, contributed reagents/materials/analysis tools, prepared figures and/or tables, authored or reviewed drafts of the paper, approved the final draft.
- Damiano Formenti conceived and designed the experiments, performed the experiments, prepared figures and/or tables, authored or reviewed drafts of the paper, approved the final draft.
- Nicola Ludwig and Marco Gargano performed the experiments, analyzed the data, contributed reagents/materials/analysis tools, authored or reviewed drafts of the paper, approved the final draft.
- Andrea Bosio performed the experiments, contributed reagents/materials/analysis tools, authored or reviewed drafts of the paper, approved the final draft.
- Ermanno Rampinini and Giampietro Alberti conceived and designed the experiments, authored or reviewed drafts of the paper, approved the final draft.

### Human Ethics

The following information was supplied relating to ethical approvals (i.e., approving body and any reference numbers):

According to the declaration of Helsinki, the study was approved by the Ethical Committee of the Università degli Studi di Milano (Approval number: 2/12).

### Data Availability

The raw data have been provided as Data S1.

### Supplemental Information

Supplemental information for this article can be found online at http://dx.doi.org/10.7717/peerj.4438#supplemental-information.

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
