# Peer review of "Bilateral asymmetry of skin temperature is not related to bilateral asymmetry of crank torque during an incremental cycling exercise to exhaustion"

_PeerJ, doi:10.7717/peerj.4438_

## Round 0.1 · original submission · Major Revisions

The three reviewers and I have found many positives in your study and its description in the written manuscript. However, all three reviewers provide some constructive criticisms that need to be considered before the manuscript can be more seriously considered for publication in PeerJ. In particular, I suggest you focus on the comments of reviewer one who listed four points at the start of their review that you need to address.

Reviewer 1 ·

Basic reporting

See comments to the authors.

Experimental design

Not assessed due to major limitation in the rationale.

Validity of the findings

Not assessed due to major limitation in the rationale.

Comments for the author

Review of article 2017:10:21073:0:0: Bilateral asymmetry of crank torque and skin temperature on incremental cycling performance

General comments
This study intended to assess the relationship between bilateral asymmetries from skin temperature and crank torque in cycling. The issue is interesting but my query is if there is sufficient theoretical rationale for this comparison. Because we have so many muscles that could add up to the total crank torque, I would expect that this relationship would not build up, which seems to be the main finding.
The abstract is well organized but would benefit from some edits (see specific comments).
The introduction is well organized as well but I see two areas that need improvement:
1- Overall rationale for assessing asymmetries in skin temperature: The argument provided by the authors is not sufficiently strong (no studies assessing this issue). My reason for questioning that is because we have resources today to measure pedal power for each leg. Therefore, no need to infer on asymmetries from skin temperature
2- Elite cyclists: Always good to see studies on elite athletes, but what is the benefit of that in this particular case? Would be good if authors develop more on that.
3- Use of the graded load test: This test has been used in the past because is valid for assessing endurance cycling performance but authors should provide a rationale why they opted for this rather than a time to exhaustion.
4- Hypothesis: In my view, the hypothesis is not supported in the introduction or by prior research. Priego et al. 2015 observed only a moderate relationship between EMG and skin temperature. Obviously, the main interest is to predict muscle forces (either via EMG or skin temperature). I am just questioning whether skin temperature would be sufficient sensitive to detect that given subcutaneous tissue seems to play a major role on that (see Priego et al. 2015 and Priego et al. 2016 - https://link.springer.com/article/10.1007/s10973-016-5971-z).

Before we assess this study further, I believe the authors have to address the aforementioned issues.

Specific comments
Title: From the article title, it is very hard to assess what is the overall aim of this study. Please revise to provide more specific details on what is intended.

Abstract:
Line 26: ‘Muscle effort’ is too broad. Please be more specific.
Lines 28-30: Graded exercise testing involve a combination of fatigue and incremented load. Please reword your aims to take this combined effect into account.
Line 30: If they are elite, please remove the word ‘competitive’.
Lines 34-35: Please state clearly what do you mean by ‘interaction’. Do you mean crank torques increased?
Lines 35-36: Similar to previous comment.

Introduction:
Lines 50-54: Please add reference to recent systematic review in this topic: http://www.tandfonline.com/doi/full/10.1080/02640414.2017.1361894
Line 62: Please add the following to your statement:
http://hrcak.srce.hr/168650?lang=en
https://ojs.ub.uni-konstanz.de/cpa/article/download/427/366
https://search.proquest.com/docview/1551927120?pq-origsite=gscholar
Line 63: Please revise given Priego et al found a moderate inverse relationship.
Lines 78-81: This is true, if we did not have easier and cheap access to power meters that provide measures of bilateral power. Please provide a stronger argument on why this assessment is critical.
Lines 83-84: I understand that this type of protocol has been used before and is reliable to assess endurance cycling performance. However, you have to develop more on why you selected this particular protocol. Keep in mind that you are dealing with a combined effect of workload and fatigue.
No comments are provided given limitations in the study rationale.

Reviewer 2 ·

Basic reporting

Abstract:
Line 31: ‘limbs’ – thigh or shank or leg.
Line 78: ‘kinetic variable’ -> kinetics/kinematics and skin temperature..

Introduction
Line 63: ‘recently it has been found’ -> Recently a relationship….. has been found.
Line 78: ‘kinetic variable’ -> kinetics/kinematics and skin temperature..


Discussion:
Line 231-237: please reword to clarify. Did Arfaoui use exactly the same 11 masters athletes? This is implied but needs to be specified. Were these masters athletes also experienced highly trained athletes?
Line 238: ‘former and latter’ -> Bertuccie et al. (2012) and Arfaoui et al. (2014)
Line 251: remove the ‘However’
Line 254-9: It is worth…. -> please reword to clarify. It is just a little convoluted to read at the moment and could be stated more simply.
Line 262: ‘point of you’ -> point of VIEW

Experimental design

Line 106 – does this include moisturiser and sunscreen, soap residue? Is it easier to say ‘was cleaned with an alcohol wipe’? if that is what happened?
Line 137: was the cyclist required to remain seated?

Validity of the findings

Should include a discussion:
- the apparent greater inter-athlete variation in temperature than force production.
- Final exhaustion AIT is negative and AIK is positive.

Comments for the author

This is a good paper that presents useful novel research. The main comments relate to expression.

·

Basic reporting

The manuscript presents an investigation on the relationship between bilateral
kinetic and skin temperature variables during a fatiguing cycling exercise.
The manuscript is mostly well written, the authors show deep knowledge in the related area and the results support the claims.
However I have some suggestions to improve the publication:
- At the abstract, refrain from using personal pronouns such as "our results...", you can simple write "the obtained results...", it is more elegant.
- Add the keyword: skin temperature
- when you present thermal asymmetry at the introduction you are missing some relevant literature such as:
Zaproudina, N., Varmavuo, V., Airaksinen, O., & Närhi, M. (2008). Reproducibility of infrared thermography measurements in healthy individuals. Physiological measurement, 29(4), 515.
Vardasca, R., Ring, E. F. J., Plassmann, P., & Jones, C. D. (2012). Thermal symmetry of the upper and lower extremities in healthy subjects. Thermology international, 22(2), 53-60.
- At methodology, nothing is referred about subject preparation in terms of intaking an heavy meal, alcohol, drugs, coffee or tea before exercise, and these factor are known for influencing skin temperature measurements.
- When the authors describe the infrared camera used, they mention thermal resolution but the value refers to NETD (Non-Equivalent Temperature Difference), which is a totally different parameter and they do not specify the traceability of the instrument.

Experimental design

In terms of methodology the sample is small to make strong conclusions, however a more refined subject preparation protocol is advised, please attend what is suggested in:
Ring, E. F. J., & Ammer, K. (2000). The technique of infrared imaging in medicine. Thermology international, 10(1), 7-14.
Ammer, K., & Ring, E. F. J. (2006). Standard procedures for infrared imaging in medicine. Biomedical Engineering Handbook, CRC Press, 1.
Ammer, K. (2008). The Glamorgan Protocol for recording and evaluation of thermal images of the human body. Thermology international, 18(4), 125-144.

Validity of the findings

The findings are in line with the presented literature background.

---

## Round 0.2 · Major Revisions

I am impressed that you have been able to address the comments from two of the three initial reviewers and that you are to be congratulated. However, the first reviewer still sees many limitations in your work, especially about the interpretation of bilateral asymmetries in skin thermography. I would therefore recommend that you look very carefully at their comments and look to take on board as many as you can in your resubmission.

Reviewer 1 ·

Basic reporting

See general comments.

Experimental design

See general comments.

Validity of the findings

See general comments.

Comments for the author

Review of article 2017:10:21073:0:0: Bilateral asymmetry of crank torque and skin temperature on incremental cycling performance

General comments
Authors provided a revised version a reply letter to my comments. Although skin temperature is a growing area in research and in sports monitoring, to me, upcoming studies (like this one) seem to be over interpreting finding from previous studies. I had the opportunity to be part of two studies using thermal infrared and we showed that this measure is not strongly related to performance or muscle activation during cycling. This leaves the question of whether skin temperature should be used as it has been in the field. This aim of this study is valid only if authors state very clearly that there is no prior expectation that asymmetries in skin temperature should be related to asymmetries in crank torque. If this is not stated clearly, the findings from this study will be over interpreted and practitioners will start using thermography to predict muscular asymmetries, which is not appropriate practice.
Therefore, my rebuttal to authors’ response leaves authors with the mission to revise their article (from top to bottom) and assume the aforementioned principles before we go through the article specifics. If authors are not willing to move this way, I cannot endorse the content of this study.

Rebuttal to author’s response
Authors provided some arguments in regards to my comments from the prior version of this study. I will pinpoint them and provide my view/comments on that:
1- Authors acknowledged that the large number of muscles in the lower limbs constrains the potential for skin thermography to accurately track asymmetries during cycling. However, they went backwards by referring to previous studies in this issue (i.e. Arfaoui et al., 2014) which does not mean that this method is reliable. In my view, authors would have a rationale in their study if they agree that asymmetries in skin temperature should not be associated with asymmetries in crank torque. This does not seem the case from what is being presented in their article. Authors should keep in mind that Priego et al. 2015 only found a moderate association between skin temperature and activation of vastus lateralis and latter a moderate association with peak power (Priego et al. 2016). This is insufficient to ascertain that skin temperature is accurate to measure muscle effort or to assess bilateral asymmetries.
2- Lack of rationale for the study: Authors moved to the absence of studies on elite athletes without providing a clear explanation on why this is important. In additional, they did not discuss the option of measuring pedal power which is more closely associated with muscle effort then skin temperature. This option should be brought clearly as the gold standard measure of asymmetries in cycling to date. In addition, the option of using musculoskeletal modeling should be discussed as an option to assess muscle asymemtries. They also escaped from the central part of the discussion by stating that their aim was to assess asymmetries in skin temperature. However, my question is what practitioners and clinicians gain in measuring skin temperature if this is not a reliable measure of muscle effort. The studies cited by authors do not add up on the story that authors are trying to tell (i.e. asymmetries in skin temperature are important). At the moment, there is no physiological background to start considering asymmetries in skin temperature as an important measure because this is not related to muscle effort. Prior publication or the lack of should not endorse future studies but only a solid rationale based on sound physiological mechanisms should be the anchor point for new studies.
3- Incomplete hypothesis: Authors again did not provide solid evidence (only reference from an opinion based article). If the aim of the authors was to assess the relationship between asymmetries in skin temperature and crank torque, a solid physiological background is needed, which in my view could only be as follows: Asymmetries in skin temperature should not be associated with crank torque because these measures do not reflect muscle effort.

No comments are provided given limitations in the study rationale.

Reviewer 2 ·

Basic reporting

This now reads much better. The figures are clear and support the prose.

Experimental design

Minutiae have now been clarified. This is well expressed/

Validity of the findings

No further comment.

Comments for the author

This is novel research and is now well expressed. It would be interesting to see in a novice group - maybe those transitioning from one form of cycling to another, or even triathletes, where there is less training effect and asymmetry may be greater. Good luck with further endeavours.

·

Basic reporting

I am happy with the authors responses and addressing to the reviewers comments.

Experimental design

I am happy with the authors responses and addressing to the reviewers comments.

Validity of the findings

I am happy with the authors responses and addressing to the reviewers comments.

Comments for the author

I am happy with the authors responses and addressing to the reviewers comments.

---

## Round 0.3 · Minor Revisions

General comments

I applaud the authors for working hard to take on board all three reviewers comments, and elderly gaining acceptance from the last two reviewers. I have reviewed your replies to that of reviewer one and feel that the following changes are still required before this manuscript can be accepted for publication.

Specific comments
line 59: This should read “cyclist to experience imbalances”.
Line 60: Please remove “indeed” from the start of the sentence.
Line 63: This should read “unfortunately only a few studies”.
Line 72: when you state significant relationships exist between variables in this manuscript it would be useful to provide the actual correlation value in the sentences. Please ensure you do it here and throughout the manuscript.
Line 83: I may have missed it but what does the abbreviation IRT stand for?
Line 87 – 89: if there were micro injuries did you expect this to increase or decrease the skin temperature asymmetry?
Line 97: provide the actual citations here instead of stating “the authors”.
Line 112 – 114: I am not convinced that you actually presented results for the first aim about the bilateral skin temperature and kinetic outcomes changes in response to maximum incremental exercise. You need to make this more explicit through the Results and Discussion, or alter the manner in which is aim is presented in this section.

---

## Round 0.4 · accepted · Accept

We thank you for the hard work you have put in to taking on board the comments of the reviewers and I. We are now happy to accept your manuscript for publication in PeerJ.